# Pd₄S/SiO₂: A Sulfur-Tolerant Palladium Catalyst for Catalytic Complete Oxidation of Methane

**Lei Ma** [1,*] , **Shiyan Yuan** [1], **Hanfei Zhu** [1], **Taotao Jiang** [1], **Xiangming Zhu** [2,*] , **Chunshan Lu** [1] **and Xiaonian Li** [1,*]

[1] Industrial Catalysis Institute, Laboratory Breeding Base of Green Chemistry-Synthesis Technology, Zhejiang University of Technology, Hangzhou 310014, China; yuanshiyan318420@163.com (S.Y.); hanfeizhu0701@foxmail.com (H.Z.); jiangtaotao2016@outlook.com (T.J.); lcszjcn@zjut.edu.cn (C.L.)

[2] Centre for Synthesis and Chemical Biology, UCD School of Chemistry, University College Dublin, Belfield, Dublin 4, Ireland

[*] Correspondence: malei@zjut.edu.cn (L.M.); xiangming.zhu@ucd.ie (X.Z.); xnli@zjut.edu.cn (X.L.); Tel.: +86-571-88320920 (L.M.)

**Abstract:** Sulfur species (e.g., H₂S or SO₂) are the natural enemies of most metal catalysts, especially palladium catalysts. The previously reported methods of improving sulfur-tolerance were to effectively defer the deactivation of palladium catalysts, but could not prevent PdO and carrier interaction between sulfur species. In this report, novel sulfur-tolerant SiO₂ supported Pd₄S catalysts (5 wt. % Pd loading) were prepared by H₂S–H₂ aqueous bubble method and applied to catalytic complete oxidation of methane. The catalysts were characterization by X-ray diffraction, Transmission electron microscopy, X-ray photoelectron Spectroscopy, temperature-programmed oxidation, and temperature-programmed desorption techniques under identical conditions. The structural characterization revealed that Pd₄S and metallic Pd⁰ were found on the surface of freshly prepared catalysts. However, Pd₄S remained stable while most of metallic Pd⁰ was converted to PdO during the oxidation reaction. When coexisting with PdO, Pd₄S not only protected PdO from sulfur poisoning, but also determined the catalytic activity. Moreover, the content of Pd₄S could be adjusted by changing H₂S concentration of H₂S–H₂ mixture. When H₂S concentration was 7 %, the Pd₄S/SiO₂ catalyst was effective in converting 96% of methane at the 400 °C and also exhibited long-term stability in the presence of 200 ppm H₂S. A Pd₄S/SiO₂ catalyst that possesses excellent sulfur-tolerance, oxidation stability, and catalytic activity has been developed for catalytic complete oxidation of methane.

**Keywords:** sulfur-tolerance; Pd₄S; catalytic oxidation of methane; sulfur poisoning

## 1. Introduction

Methane as the main component of natural gas is playing an increasingly important role in the global energy structure [1,2]. Effective catalytic complete oxidation of CH₄ can improve combustion efficiency and reduce air pollutants, such as CO, NOₓ, and unburned hydrocarbon [3–8]. It has therefore found great applications in modern industry, such as catalytic exhaust converters aimed to reduce methane emission and catalytic gas turbine combustors designed to combust fuel under mild conditions [9,10]. Supported PdO catalysts have shown excellent catalytic property in methane oxidation, and currently are under extensive study [11–18]. However, once sulfur species (e.g., H₂S or SO₂) are present in the reaction atmosphere, the poisoning of PdO catalyst which would lead to inactive PdO-SOₓ is irreversible and the activity of the catalyst cannot be recovered at relatively low temperature [19–25].

Hence, many efforts have been devoted in the past decade to enhance sulfur resistance of supported PdO catalysts. Currently there are two primary approaches to improve the performance of palladium

catalysts against sulfur poisoning: the first is to use alkaline carriers (e.g., $\gamma$-$Al_2O_3$) and enhance their adsorption ability toward acidic sulfur species to protect the active phase of PdO [23,26,27], and the second is to introduce an extra active ingredient to form a catalyst system with dual activity, such as Pd–Pt complex, to avoid sulfur poisoning [24,28]. However, these methods can only defer the deactivation of palladium catalysts as neither the alkaline carrier nor the extra active ingredient can prevent the interaction between PdO and sulfur species. In other words, sulfur poisons would eventually encroach PdO and make it inactive. As such, these catalyst systems did not really possess the ability of sulfur-tolerance.

The research on sulfur-tolerant palladium catalyst system in methane catalytic oxidation reaction is rare and lack of substantial progress in the literature [20,26–28]. Therefore, development of a reliable sulfur-tolerant catalyst system seems to be of great importance. In view of the great potential of catalytic oxidation of methane, we decided to pursue a new catalytic system that could be of high tolerance of sulfur poisons. We speculated $Pd_xS_y$ in combination with acidic carrier would be an ideal system for methane oxidation based on the following two points. First, it would be the best scenario for sulfur-tolerant catalysts that neither the support nor the palladium active phase itself could interact with sulfur species. $Pd_xS_y$ and acidic carrier such as $SiO_2$ would likely meet the requirement. Second, seeing that sulfur and oxygen are located in the adjacent position of the same main group in the periodic table, i.e., the chalcogen group, the designed $Pd_xS_y$ might possess similar catalytic activity to PdO.

In the literature, the majority of $Pd_xS_y$ ($Pd_4S$, $Pd_3S$, $Pd_{2.8}S$, $Pd_{16}S_7$, PdS or $PdS_2$, etc.) were prepared by gas sulfuration with $H_2S$–$H_2$ [25,29–35]. Herein, we would like to present a new procedure for the preparation of $Pd_xS_y$/$SiO_2$ catalysts via aqueous bubble sulfuration of Pd/$SiO_2$ with $H_2S$–$H_2$ with varied $H_2S$ concentration. Compared with the gas sulfuration, the aqueous bubble method was performed in much milder conditions. Besides the sulfur resistance and catalytic activity, the stability of $Pd_xS_y$/$SiO_2$ catalyst was the other focus of our investigation.

## 2. Results and Discussion

### 2.1. Fresh Catalysts

#### 2.1.1. Catalyst Characterization

Freshly prepared catalysts were first tested by X-ray diffraction (XRD) as showed in Figure 1. The freshly prepared catalyst is named (fV), where V represents $H_2S$ concentration. No other palladium species were found except for metallic $Pd^0$. The sulfuration of Pd to form $Pd_4S$ under $H_2S$–$H_2$ atmosphere was via the reaction: $4Pd + H_2S \leftrightarrow Pd_4S + H_2$ [36–40]. This reversible reaction meant that metallic $Pd^0$ could not be fully sulfurized to $Pd_4S$. Therefore, it was reasonable to detect metallic $Pd^0$ on the surface of freshly sulfurized catalyst. However, it was worth noting that the intensity of the diffraction peak of metallic $Pd^0$ changed regularly with the increase of $H_2S$ concentration. The average particle size of metallic $Pd^0$ calculated by Scherrer formula was further shown in Table 1. With the increase of $H_2S$ concentration, the change of average particle size of metallic $Pd^0$ could be divided into three stages. They were gradually reduced (0 → 5%), stable (5% → 7%), and increased (7% → 10%). These results indicated the Pd/$SiO_2$ precursor had been changed by sulfuration.

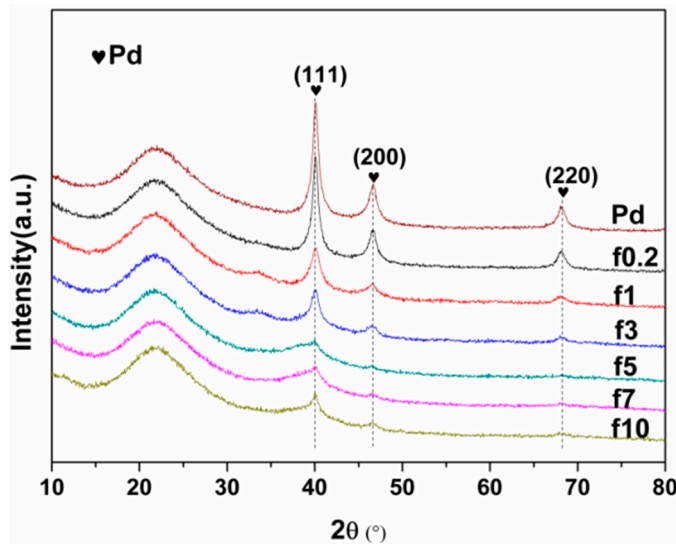

**Figure 1.** X-ray diffraction (XRD) patterns of freshly prepared catalysts sulfurized with different $H_2S$ concentration.

**Table 1.** Average particles size of freshly prepared catalysts sulfurized with different $H_2S$ concentration.

| Catalyst | FWHM | Size (nm) |
|---|---|---|
| Pd | 0.761 | 12.2 |
| f0.2 | 0.803 | 11.5 |
| f1 | 1.328 | 6.9 |
| f3 | 1.297 | 6.7 |
| f5 | 3.329 | 2.6 |
| f7 | 3.262 | 2.6 |
| f10 | 2.163 | 4.0 |

FWHM: full-width at half-maximum

Since XRD characterization could not directly provide evidence of the existence of $Pd_xS_y$, the freshly prepared catalysts were further characterized by high resolution transmission electron microscopy (HR-TEM) and illustrated in Figure 2. The lattice spacing distance of Pd species particles on freshly prepared catalyst are 0.225 nm and 0.245 nm, corresponding to the Pd(111) and $Pd_4S$(102), respectively [JCPDS No. 73-1387, JCPDS No. 46-1043]. In addition to the metallic Pd(111), the $Pd_4S$(102) was found on the surface of all the freshly prepared catalysts. The literature on the preparation of $Pd_xS_y$ under $H_2S$-$H_2$ atmosphere showed that the $Pd_4S$ with the highest proportion of Pd/S was the first $Pd_xS_y$ species [38,41]. Therefore, it was reasonable to firstly form the $Pd_4S$ under the condition of 60 °C aqueous bubble sulfuration.

The subsequent characterization by X-ray photoelectron spectroscopy (XPS) could provide further evidence for the existence of $Pd_xS_y$, as shown in Figure 3. After curve fitting analysis, the $S2p_{3/2}$ of freshly prepared catalyst could be deconvoluted into a large peak at 168.5 eV and a small peak at 165.1 eV, which corresponded to $S^0$ [42] and $Pd_4S$ [42–44], respectively. With the increase of $H_2S$ concentration, the XPS peak intensity of sulfur species increased gradually. At the same time, the $Pd3d_{5/2}$ could be deconvoluted into two peaks at ~335.6 eV and ~337.5 eV which corresponded to metallic $Pd^0$ [45,46] and $Pd_4S$ [47,48], respectively. This indicated that $Pd_4S$ and metallic $Pd^0$ were the primary palladium species on the freshly prepared catalysts. These results are in good agreement with the characterization results of XRD and HR-TEM. If the $Pd_4S$ had similar properties to PdO, the amount of $Pd_4S$ would be the key factor influencing the performance of the catalyst. The XPS data (Figure 3) were further analyzed to provide Table 2 that listed the composition ratio of palladium species of freshly prepared catalysts. It could be noted that the sulfuration process was the transition of metallic $Pd^0$ to $Pd_4S$. With the increase of $H_2S$ concentration, the change of $Pd_4S$ ratio could also be divided into

three stages. They were gradually increased (0 → 5%), stable (5% → 7%), and reduced (7% → 10%). Variation of metallic $Pd^0$ content was opposite to that of $Pd_4S$. This variation was entirely consistent with the average particle size of metallic $Pd^0$. It was obvious that the variation of the average particle size of metallic $Pd^0$ was caused by the change of metallic $Pd^0$ content on the catalyst surface.

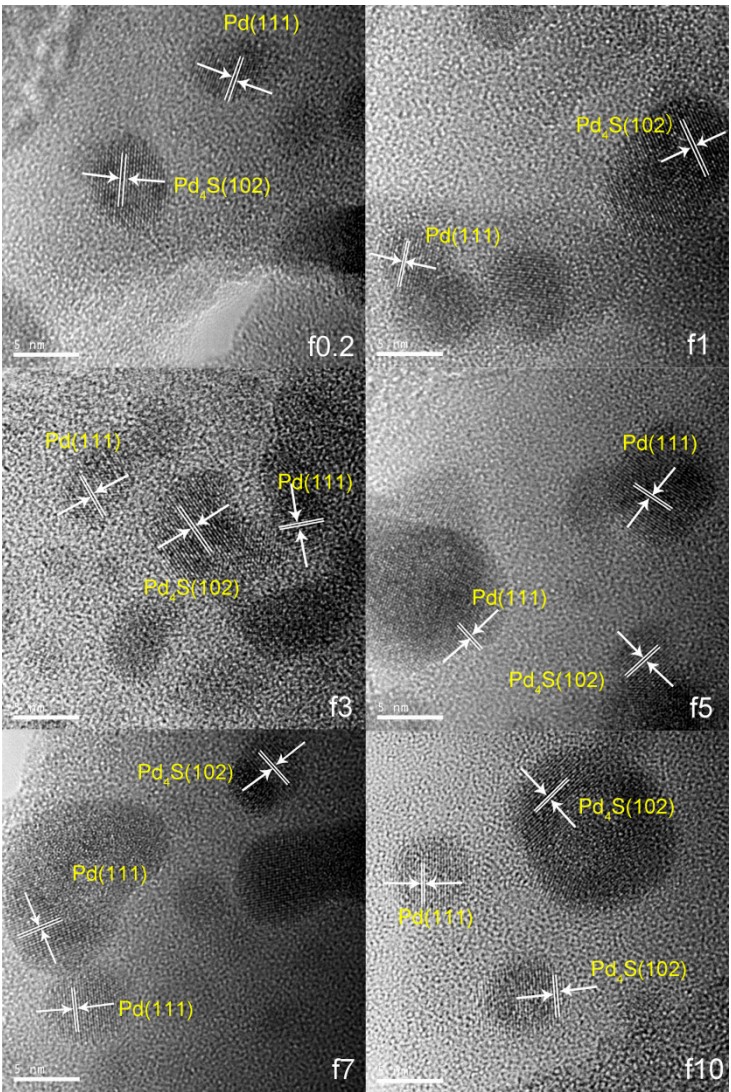

**Figure 2.** High resolution transmission electron microscopy (HR-TEM) images of freshly prepared catalysts with different $H_2S$ concentration.

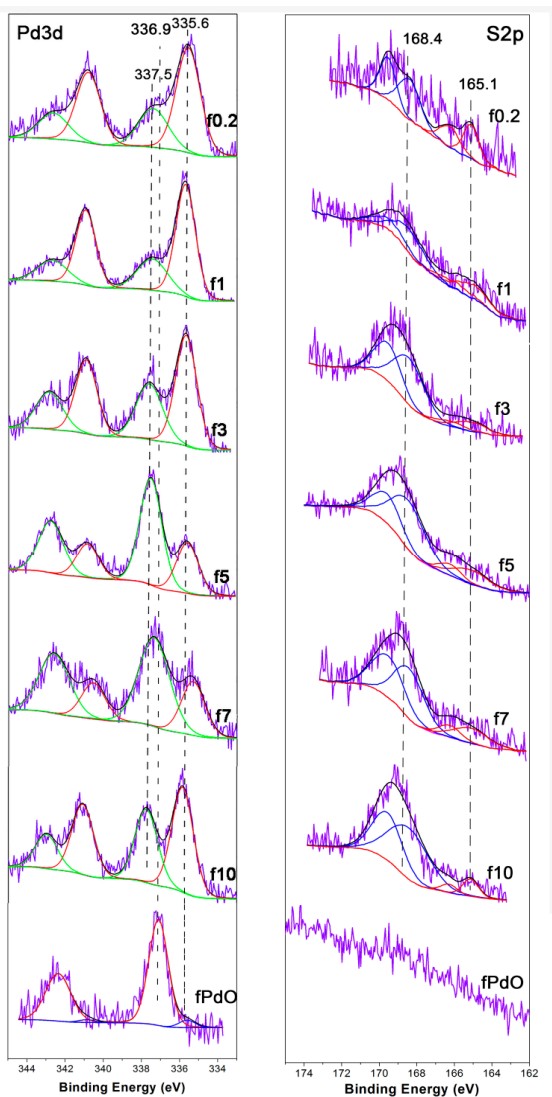

**Figure 3.** X-ray photoelectron spectroscopy (XPS) Pd3d and S2p of freshly prepared catalysts with different $H_2S$ concentration.

**Table 2.** Palladium species content of freshly prepared catalysts with different $H_2S$ concentration [a].

| Catalyst | Composition Ratio of Palladium Species (%) | |
| :---: | :---: | :---: |
| | $Pd^0$ | $Pd_4S$ |
| f0.2 | 68.9 | 31.1 |
| f1 | 68.6 | 31.4 |
| f3 | 61.6 | 38.4 |
| f5 | 33.4 | 66.6 |
| f7 | 32.1 | 67.9 |
| f10 | 57.9 | 42.1 |

[a]: Calculated by Pd fitted peak area by XPS.

### 2.1.2. Activity Studies

In order to test whether the $Pd_4S$ possessed the desired sulfur-tolerance and catalytic activity, its catalytic property for methane oxidation in the presence of 200 ppm $H_2S$ was then examined by measuring methane conversion against the reaction time at different reaction temperatures. As shown in Figure 4, when the temperature was 500 °C, the presence of $H_2S$ had no impact on all the catalysts. However, once the reaction temperature dropped to 400 °C, the compared PdO/SiO$_2$ catalyst was

deactivated very rapidly. We estimated that at this temperature sulfur species could turn the active PdO into inactive PdO–SO$_x$ species, which could then hardly decompose at low temperature [23,24,49]. In contrast, under the same conditions, all the catalysts prepared by our own procedure did not show any deactivation at 400 °C. This clearly stated that the combination of Pd$_4$S and SiO$_2$ was effective in resisting sulfur poisoning. At the same time, the activity of the catalyst with 7% H$_2$S concentration was even equal to that of PdO catalyst. This further stated that the combination of Pd$_4$S and SiO$_2$ had high enough catalytic activity. Furthermore, it was found that the methane conversion of all catalysts decreased to less than 10% at 370 °C. This dramatic decrease in catalytic activity was not due to sulfur poisoning, but the high reaction temperature required for methane activation [50–52]. Therefore, it was best to set the reaction temperature at 400 °C. This temperature could not only reflect the sulfur-tolerance, but also compare the catalytic activity.

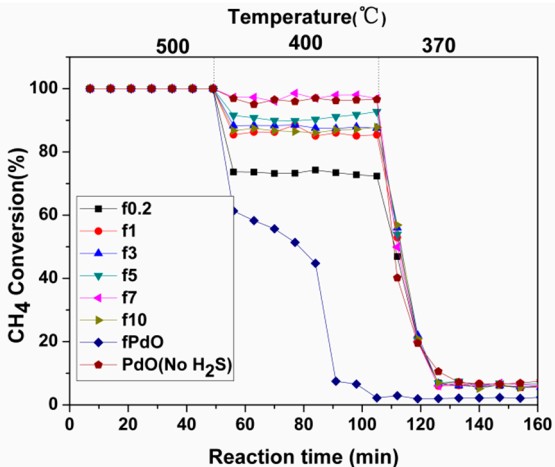

**Figure 4.** Catalytic performance of freshly prepared catalysts with different H$_2$S concentration. Gas composition: CH$_4$ (v% = 2%), O$_2$ (v% = 8%), H$_2$S (v% = 0.02%), and N$_2$ (v% = 89.8%); Flow rate = 200 mL/min; and GHSV (gas hourly space velocity) = 60000 mL/(g·h).

Meanwhile, it should be pointed out here that although all the sulfurized catalysts exhibited the similar sulfur-tolerance, the catalytic activity at 400 °C was not same. As the H$_2$S concentration increased from 0.2% to 7%, the catalytic activity increased gradually. However, when the H$_2$S concentration increased to 10%, the catalytic activity decreased. This variation was very similar to the variation of Pd$_4$S content on the surface of freshly prepared catalyst. However, we could not simply assume that catalytic activity depended solely on Pd$_4$S. The reason was that the stability of Pd$_4$S and metallic Pd$^0$ on the surface of freshly prepared catalyst was still undiscovered under high reaction temperature and presence of oxygen. Under such reaction condition, metallic Pd$^0$ was liable to be oxidized to PdO. Therefore, palladium species on the surface of freshly prepared catalyst might not be the true palladium species involved in the reaction. Next, we would characterize the used catalyst to determine the actual palladium species on the catalyst surface.

## 2.2. Used Catalyst

### 2.2.1. Catalyst Characterization

The XRD patterns of used catalysts were given in Figure 5. The used catalyst is named (uV), where V represents H$_2$S concentration. The characteristic diffraction peak attributed to sulfur species was found on the compared PdO/SiO$_2$ catalyst. However, no diffraction peaks attributed to sulfur species were found in all sulfurized catalysts after the reaction. The reason should be that the sulfurized catalyst had the ability to resist sulfur poisoning. Not surprisingly, almost all the diffraction peaks observed in the sulfurized catalyst were attributed to PdO after the reaction. A weak diffraction peak of metallic Pd$^0$ could be observed only when H$_2$S concentration was 0.2%. This implied that most of

the metallic $Pd^0$ on the freshly prepared catalyst had been oxidized to PdO under reaction conditions. However, the above results did not show whether the most important $Pd_4S$ remains stable during the reaction and this needed to be identified by other characterization.

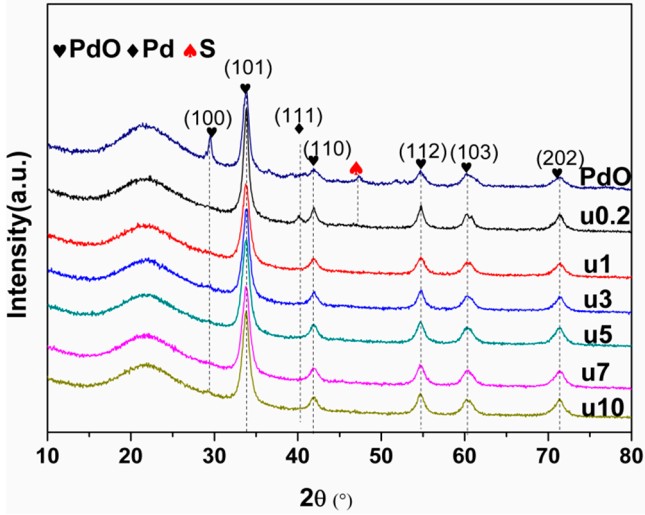

**Figure 5.** XRD patterns of used catalysts with different $H_2S$ concentration.

The presence of PdO confirmed our previous speculation that the palladium species on the surface of freshly prepared catalysts were not entirely stable during the vigorous oxidation reaction. However, XRD characterization could only reveal that PdO species were formed during the reaction of freshly prepared catalysts. However, whether the essential $Pd_4S$ species could stably exist in large quantities would be the main factor affecting the sulfur-tolerance of the catalyst. Figure 6 shows HR-TEM images of the used catalysts. It could be found that the palladium species on the surface of the used catalyst were more complicated than the freshly prepared catalyst. Figure 6 clearly indicates that in addition to $Pd^0(111)$, PdO(101) and PdO(110) also appeared in large numbers on the surface of all the used catalysts [JCPDS No.06-0515]. At the same time, the most interesting thing was that $Pd_4S(102)$ species could be found on all catalyst surfaces. This indicated that the $Pd_4S$ species could remain stable during the high temperature and high oxygen reaction of methane oxidation.

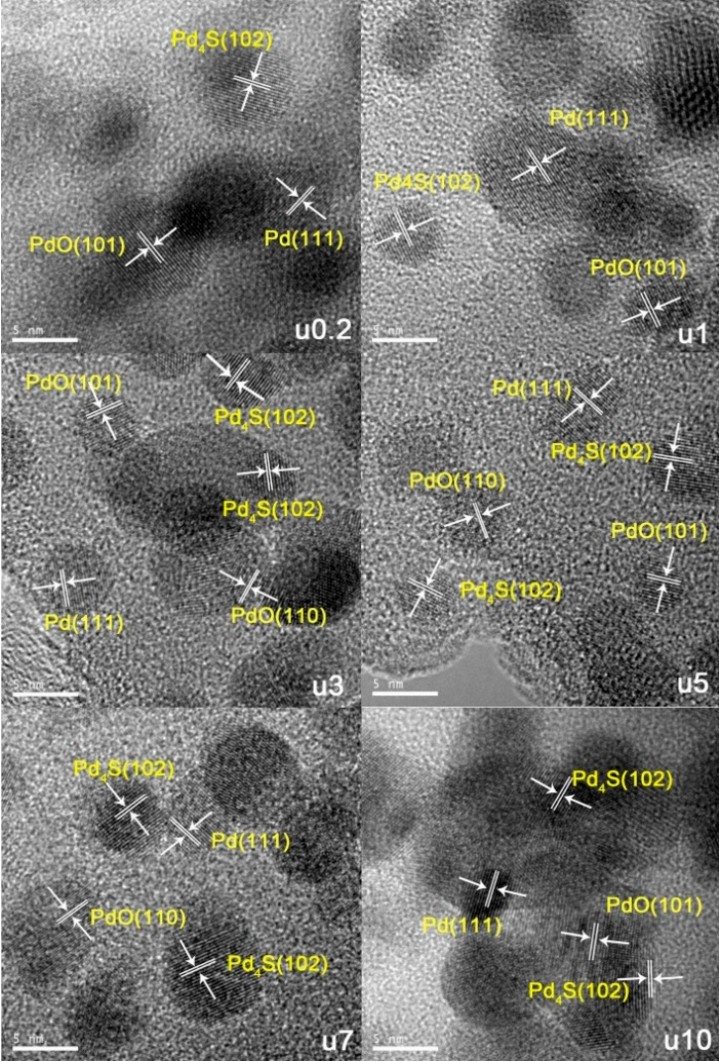

**Figure 6.** HR-TEM images of the used catalysts with different $H_2S$ concentration.

Owing to the lack of data on the relative content of PdO on the surface of used catalyst, when $Pd_4S$ and PdO coexist, we could not distinguish whether $Pd_4S$ and PdO affect the catalytic activity alone or in combination. Therefore, used catalysts were further analyzed by XPS characterization, as shown in Figure 7. After curve fitting analysis, the $Pd3d_{5/2}$ could be deconvoluted into three peaks at ~337.5 eV, ~336.9 eV, and ~335.6 eV, which were assigned to $Pd_4S$ [47,48], PdO [48], and metallic $Pd^0$ [45,46], respectively. The above characterization results revealed the fact that under the reaction conditions, the actual palladium species on the used catalyst surface should be $Pd_4S$, PdO, and metal $Pd^0$. At the same time, the S2p peak attributed to $S^0$ was not observed (see Figure 3), only the very weak S2p peak attributed to $Pd_4S$ could be observed. In contrast, the Pd3d peak at ~337.3 eV and S2p peak at 168.1 eV of used $PdO/SiO_2$ could be attributed to $PdO–SO_x$. This comparison results indicated that $Pd_4S$ could protect other palladium species from sulfur poisoning.

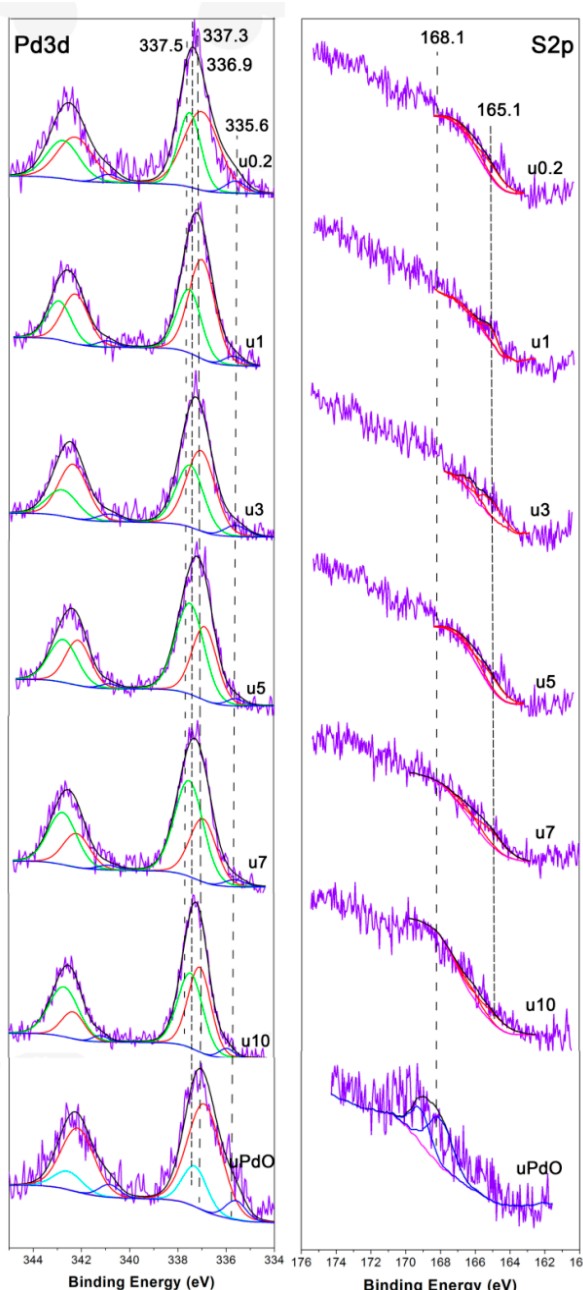

**Figure 7.** XPS Pd*3d* and S2p of the used catalysts with different $H_2S$ concentration.

Table 3 listed the composition ratio of palladium species of used catalysts. Compared with the relative content of palladium species on the surface of fresh catalyst (Table 1), it could be noted that a large number of PdO species appeared on the surface of the used catalyst, while the $Pd^0$ content of the metal decreased significantly. This was clearly the oxidation of metal $Pd^0$ to PdO during the reaction. At the same time, the relative content of $Pd_4S$ was basically stable during the reaction. This indicated that the PdO formed during the reaction was mainly derived from the metal $Pd^0$, but not derived from $Pd_4S$. Temperature-programmed Oxidation (TPO) of fresh $Pd/SiO_2$, f7 and u7 catalysts with the same mass was investigated by thermogravimetric analysis. The oxidation stability of the catalyst was examined by recording the mass change in the process of air oxidation. The results are presented in Figure 8. Three samples had a similar dehydration process before 100 °C. However, with the increase of temperature, the three samples showed completely different changes. Freshly prepared $Pd/SiO_2$ had the greatest mass increase, which should be related to the oxidation of metallic $Pd^0$ to PdO. The mass

increase of f7 with the same mass was significantly smaller than that of freshly prepared Pd/SiO$_2$, while u7 with the same mass had no significant mass increase. f7 was composed of metallic Pd$^0$ and Pd$_4$S, while u7 was mainly composed of PdO and Pd$_4$S. This stated that the mass increase of f7 was mainly related to the oxidation of metallic Pd$^0$, while Pd$_4$S maintained sufficient oxidation stability even under temperature higher than 500 °C. This meant that Pd$_4$S could not be oxidized into palladium oxide nor into palladium sulfate. Therefore, it could be possible to conclude that in the low-temperature catalytic oxidation of methane, Pd$_4$S had sufficient stability to resist the oxidation of oxygen.

**Table 3.** Palladium species content of used catalysts with different H$_2$S concentration [a].

| Catalyst | Composition Ratio of Palladium Species (%) | | |
|---|---|---|---|
| | **Pd** | **PdO** | **Pd$_4$S** |
| u0.2 | 5.3 | 59.5 | 35.2 |
| u1 | 4.0 | 60.4 | 35.6 |
| u3 | 5.9 | 54.2 | 39.9 |
| u5 | 2.7 | 40.4 | 56.9 |
| u7 | 3.1 | 36.6 | 60.3 |
| u10 | 3.6 | 48.9 | 47.5 |

[a]: Calculated by Pd fitted peak area by XPS.

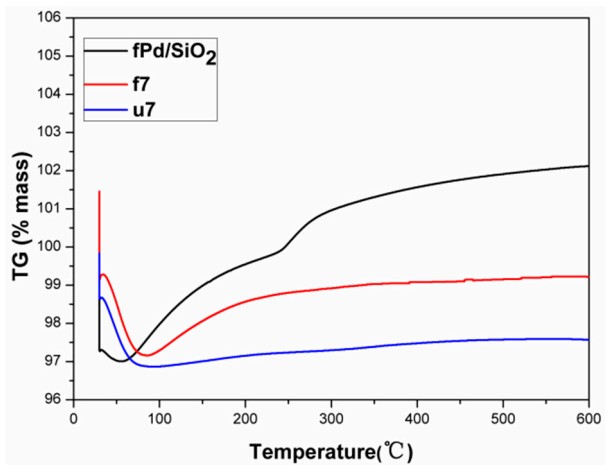

**Figure 8.** Temperature-programmed Oxidation (TPO) profiles of the (u7, f7) catalyst and freshly prepared Pd/SiO$_2$.

### 2.2.2. Activity Studies

Figure 9 covered the evaluation of the used catalysts and the evaluation method was completely consistent with the freshly prepared catalysts. To our delight, the sulfur-tolerance and catalytic activity of each used catalyst was almost unchanged with the fresh catalyst at 400 °C, which indicated that the composition of the catalyst had remained stable after the initial evaluation. The previous characterization results had confirmed that Pd$_4$S, PdO, and metal Pd$^0$ were the three palladium species actually involved in the reaction. In the case that PdO and metal Pd$^0$ did not have sulfur-tolerance, it was the existence of Pd$_4$S that gave the palladium active component the ability to resist the poisoning of sulfur species. Moreover, to our surprise, Pd$_4$S was not merely resistant to sulfur itself, its presence could even protect PdO and metal Pd$^0$ from the poisoning of sulfur species.

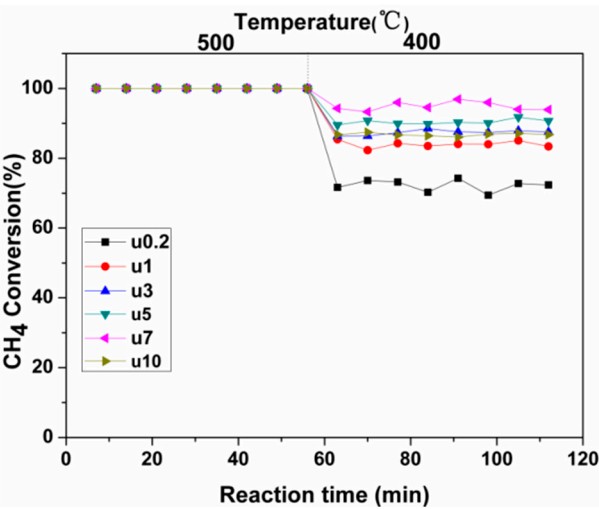

**Figure 9.** Catalytic performance of used catalysts with different $H_2S$ concentration. Gas composition: $CH_4$ (v% = 2%), $O_2$ (v% = 8%), $H_2S$ (v% = 0.02%), and $N_2$ (v% = 8 9.8%); Flow rate: 200 mL/min; and GHSV = 60000 mL/(g·h).

The sulfur-tolerance of the catalyst depended on $Pd_4S$, but it remained to be confirmed if catalytic activity was related to the type of palladium species. To this end, we correlated the relative amounts of $Pd_4S$ and PdO species on the used catalyst surface with the catalytic activity of the used catalyst. The secondary reaction performance of the catalyst was selected at 400 °C and the results are shown in Figure 10. It could be noted that the variation of catalytic activity at 400 °C was closely related to the change of $Pd_4S$ content with $H_2S$ concentration increase. As the $H_2S$ concentration increased from 0.2% to 7%, the relative content of $Pd_4S$ and the catalytic activity of the used catalyst gradually increased. Compared with content change of $(Pd_4S + PdO)$ and PdO, the content of $(Pd_4S + PdO)$ did not change with the concentration of $H_2S$, but the change of PdO content was opposite to the catalytic activity of the used catalyst. The above results showed clearly that not only the sulfur-tolerance, but also the catalytic activity relied completely on $Pd_4S$. Because the coexistence of PdO did not play any decisive role, it was completely reasonable to use $Pd_4S/SiO_2$ as the catalyst prepared by the aqueous bubble sulfuration method. The consequences of 72-hour stability test of the u7 catalyst with and without $H_2S$ are shown Figure 11. It could be found that the catalyst could maintain long-term stability regardless of the presence of $H_2S$ in the reaction atmosphere.

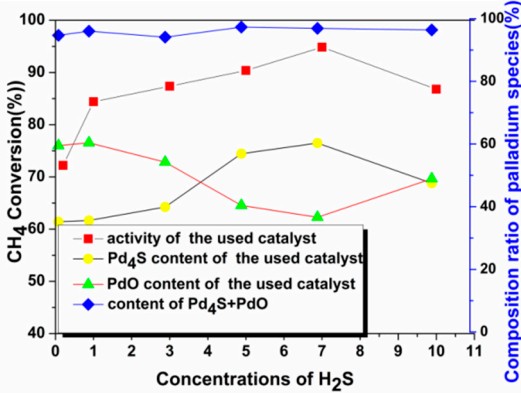

**Figure 10.** Relationship profile between composition ratio of $Pd_4S$ and PdO on used catalysts and second evaluation of methane conversion. The composition ratio of palladium species was calculated through Pd fitted peak area by XPS.

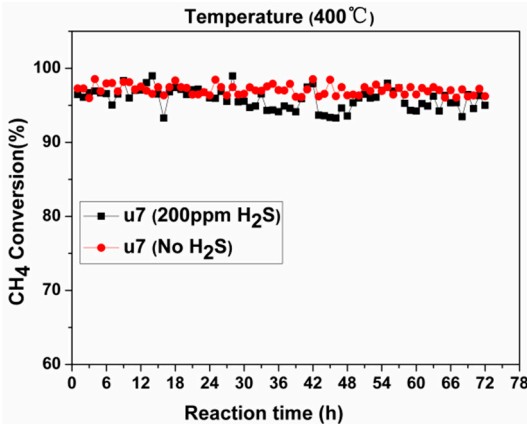

**Figure 11.** Long-term stability test of the u7 catalyst with and without $H_2S$. Gas composition: $CH_4$ (v% = 2%), $O_2$ (v% = 8%), $H_2S$ (v% = 0.02%), and $N_2$ (v% = 89.8%); Flow rate: 200 mL/min; and GHSV = 60000 mL/(g·h).

## 2.3. Mechanism of Sulfur-Tolerance

Next, we turned to investigate the cause of sulfur-tolerance of the $Pd_4S/SiO_2$ using the technique of hydrogen sulfide temperature-programmed desorption ($H_2S$-TPD). The u7 catalyst and the freshly prepared $PdO/SiO_2$ catalyst were chosen for comparison and the results are presented in Figure 12. $SO_2$ and $H_2S$ were the primary desorption species detected in the study. The $PdO/SiO_2$ catalyst released $SO_2$ in relatively large quantity, which suggested that PdO could readily assimilate $H_2S$ to form $PdO–SO_x$. As a result, $PdO/SiO_2$ catalyst was poisoned and deactivated by $H_2S$. In contrast, desorption products were barely detected from the u7 catalyst. Seeing that $Pd_4S$ was the only discrepancy between the two catalysts, i.e., the presence of $Pd_4S$ in u7 and its absence in $PdO/SiO_2$, we were convinced that $Pd_4S$ played a vital role in the sulfur-tolerance of u7 catalyst, as depicted in Scheme 1. Aside from not adsorbing $H_2S$, $Pd_4S$ also prevented PdO from adsorbing $H_2S$ onto its surface. Therefore, it was the $Pd_4S$ that blocked the adsorption of $H_2S$ and endowed the catalyst with sulfur-tolerance.

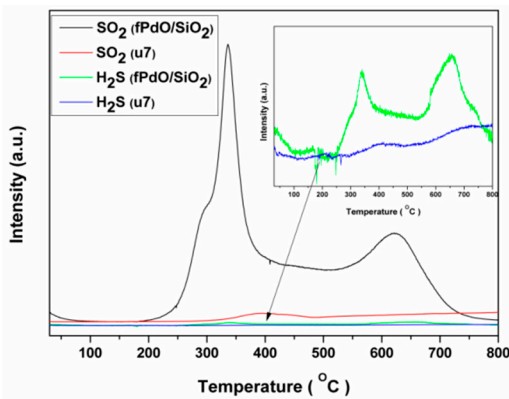

**Figure 12.** Hydrogen sulfide temperature-programmed desorption ($H_2S$-TPD)profiles of the u7 catalyst and freshly prepared $PdO/SiO_2$ catalyst.

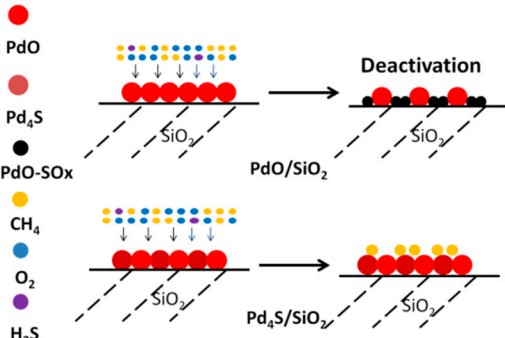

**Scheme 1.** Schematic diagram of methane catalytic oxidation on the surface of $Pd_4S/SiO_2$ and $PdO/SiO_2$ in presence of $H_2S$.

## 3. Materials and Methods

### 3.1. Catalyst Preparation

The $PdO/SiO_2$ catalyst was prepared by isovolumetric impregnation. Typically, weigh a proper quantity of 40~60 mesh $SiO_2$ (M = 10 g Vg = 0.94 $cm^3$/g, $R_d$ = 10.6 nm, S = 353.0028 $m^2$/g, Qingdao Baisha Catalyst Factory, Qingdao, China) was dispersed in palladium acetate (Aladdin Industrial Corporation, Shanghai, China) aqueous solution ($V_{Pd}$ = 0.05 g/mL) overnight. Then the sample was dried in an oven at 110 °C for 4 h and calcined at 500 °C for 4 h in the air. The theoretical loading of Pd was 5 wt. %.

The $PdO/SiO_2$ was first reduced to $Pd/SiO_2$ by $H_2$ at 500 °C for 1h. $Pd_xS_y/SiO_2$ catalyst was prepared by using above $Pd/SiO_2$ as precursor. 2.0 g $Pd/SiO_2$ and 100 ml deionized water were stirred at 350 rpm in a three-necked flask. Adjust the flow rate of $H_2S$ and $H_2$ by flow controller (D07-19B, Beijing Sevenstart Electronics Co. Ltd, Beijing, China) to configure different concentrations of $H_2S/H_2$ mixture. Then 30 mL/min of $H_2S$–$H_2$ ($V_{H2S}$% = 0.2, 1, 3, 5, 7, 10) was fed into the suspension at 60 °C for 1 h. Afterwards, the sample was filtered and rinsed with distilled water till neutral. Finally, the sample was dried at 110 °C for 4 h. The freshly prepared catalyst is named (fV), and the used catalyst is named (uV), where V represents $H_2S$ concentration.

### 3.2. Catalyst Characterization

X-Ray Diffraction (XRD) was carried out on a Thermo ARL X'TRA diffractometer (PNAlytical Co. Holland) using Cu K-a radiation (45 kV, 40 mA). Average particle size was determined from XRD measurements using the Scherrer formula:

$$d = \frac{K\lambda}{\beta \cos \theta},$$

where d is the average particle size(nm), K is the Scherrer constant and the diffraction angle is denoted θ, λ = 0.154 nm stands for the wavelength of Cu K-a radiation, β denotes the full-width at half-maximum (FWHM) of diffraction peak.

High Resolution Transmission Electron Microscopy (HR-TEM) images were taken by a Philips-FEI Tecnai G2 F30 S-Twin transmission electron microscope operated at 300 kV (Philips-FEI Co. Holland).

X-Ray Photoelectron Spectroscopy (XPS) was measured on a Thermo ESCALAB 250 Axis Ultra (KRATOS, Kanagawa, Japan) using a monochromated Al Kα X-ray source (hv = 1485.6 eV) with a fixed analyzer pass energy of 80 eV. The binding energy values were referenced to the Si 2p as internal standard (Si 2p = 103.4 eV) and the maximum deviation value was 2.6 eV in the sample. After subtraction of the Shirley-type background, the core-level spectra were decomposed into their component with mixed Gaussian-Lorentzan lines by a non-linear least squares curve fitting procedure, using the public software package XPSPEAK4.1. The corresponding atomic ratio in different chemical

environment was obtained from the fitted XPS spectra of Pd 3d and S 2p. The XPS profiles were fitted by the software named "XPS peak". Then the fitting peak area was used to calculate the proportion of different species.

Temperature-programmed Oxidation (TPO) was performed on a NETZSCH STA 409 PC/PG instrument (NETZSCH Co. Selbc, Germany) The oxidation stability of the catalysts was investigated by recording the mass change in the process of air oxidation. TPO was performed by heating 0.01 g sample from room temperature to 600 °C in air atmosphere with a heating rate of 5 °C/min and a flow rate of 40 mL/min.

Temperature-programmed Desorption (TPD) of hydrogen sulfide experiments were performed in a self-made tubular quartz reactor (5mm i.d.). The sample (0.2 g) was first swept at 110 °C for 1 h using pure He and cooled to room temperature in the same atmosphere. Then the sample was in situ treated with $H_2S$ (0.2% in $N_2$) at a flow rate of 30 mL/min for 0.5 h and swept 1 h to remove physisorbed and/or weakly bound species. TPD was performed by heating the sample from room temperature to 800 °C with constant increase of 5 °C/min in pure He. The TPD spectra were recorded by a quadrupole mass spectrometer (QMS 200 Omnistar, Pfeiffer Co. Germany).

### 3.3. Evaluation of Catalysts

The catalytic activity of the freshly prepared and used catalysts was tested with 0.2 g sample in a self-made continuous fixed bed reactor (8 mm i.d.) at atmospheric pressure. Gases consisting of $CH_4$ (v% = 2 %), $O_2$ (v% = 8 %), $H_2S$ (v% = 0.02 %), and $N_2$ were fed through the flow controller (D07-19B, Beijing Sevenstart Electronics Co. Ltd., Beijing, China) at 200 mL/min and used in all the experiments. Finally, the temperature control controller (AI808PK1L2, XiaMen YuDian Automation Technology Co., Ltd., XiaMen, China) is used to control different reaction temperatures. The effluent gases were sampled and simultaneously analyzed online by gas chromatographs (GC-9790, Zhejiang Fuli Analytical Instruments Corp., Hangzhou, China). Exhaust gases were analyzed using a PoraPak Q column and a thermal conductivity detector (TCD). Methane conversions were calculated for outlet $CO_2$ concentrations.

## 4. Conclusions

Through the characterization of freshly prepared catalyst and used catalyst, we confirmed that $Pd_4S$, PdO, and metallic $Pd^0$ were the actual palladium species involved in the oxidation of methane. Sulfur-tolerance and catalytic activity was completely dependent on $Pd_4S$, and independent on other palladium species. Moreover, $Pd_4S$ had sufficient oxidation stability under reaction conditions. Therefore, even in the presence of PdO, there are enough reasons to represent the catalyst with $Pd_4S/SiO_2$. In summary, we have developed a powerful sulfur-tolerant catalyst for methane oxidation by incorporating $Pd_4S$ into $SiO_2$, and to the best of our knowledge, this is the first report on Pd catalysts with such property. In view of the excellent sulfur-tolerant property, the excellent oxidation stability and the high catalytic activity, the $Pd_4S/SiO_2$ catalyst will definitely find valuable and versatile use in catalytic chemistry.

At the same time, as the $Pd_xS_y$ species with the highest proportion of Pd/S, we cannot determine whether the outstanding performance of $Pd_4S$ in methane catalytic oxidation is only a special case. Furthermore, sulfur-tolerant mechanism of $Pd_4S$ needs further theoretical research.

**Author Contributions:** Conceptualization, L.M.; Data curation, S.Y. and H.Z.; Formal analysis, L.M., S.Y., and H.Z.; Funding acquisition, L.M. and C.L.; Investigation, T.J.; Methodology, X.Z.; Project administration, L.M. and X.L.; Writing—original draft, S.Y.; Writing—review & editing, L.M. and X.Z.

**Funding:** This work is financially supported by National Natural Science Foundation of China (Grant No.21473159, 21476208).

**Conflicts of Interest:** The authors declare no conflict of interest.

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
