# Peer review of "Pd4S/SiO2: A Sulfur-Tolerant Palladium Catalyst for Catalytic Complete Oxidation of Methane"

_catalysts, doi:10.3390/catal9050410_

Round 1

Reviewer 1 Report

A file on Comments and suggestions is attached 

Author Response

Dear Editor:

We would like to thank the editor for giving us a chance to revise the paper and the reviewers for their constructive suggestions to improve the quality of the paper. Here we submit a revision with the title Pd4S/SiO2: A sulfur-tolerant palladium catalyst for catalytic complete oxidation of methane” (ID: Catalysts-489207), which has been modified according to the reviewers’ suggestions. All detailed revisions are listed in response to comments. We hope this version can satisfy the quality of Catalysts.

Sincerely yours,

Lei Ma

--------------------------------------------------------------------------------------------------

The following is a point-to-point response to the reviewers’ comments.

REVIEWER REPORT(S):
Referee: 1

1. General comment about references: The references are not up to date; many good articles have been published during 2018-2019 on methane oxidation. Some citations are given below. The authors should include more current research and two recent review articles in the introduction and also in the body of the text:

l  Banerjee, A. C., Golub, K. W., Abdul, Md. H., Billor, M.Z. (2019). Comparative study of the characteristics and activities of Pd//-Al2O3 catalysts prepared by Vortex and Incipient Wetness Methods. Catalysts, 9(4), 336; https://doi.org/10.3390/catal9040336.

l  Banerjee, A.C., McGuire, M. M., Lawnick, O., Bozack, M.J. (2018). Low-temperature activity and PdO/PdOx transition in methane combustion by a PdO-PdOx/y-Al2O3 catalyst. Catalysts, 8(7), 266; https://doi.org/10.3390/catal8070266.

l  Chen, J.; Arandiyan, H.; Gao, X.; Li, J. Recent Advances in Catalysts for Methane Combustion. Catal. Surv. from Asia 2015, 140–171.

l  Monai, M.; Montini, T.; Gorte, R.J.; Fornasiero, P. Catalytic Oxidation of Methane: Pd and Beyond. Eur. J. Inorg. Chem. 2018, 2018, 2884–2893.

l  Jiang, L.; Zheng, Y.; Chen, X.; Xiao, Y.; Cai, G.; Zheng, Y.; Zhang, Y.; Huang, F. Catalytic Activity and Stability over Nanorod-Like Ordered Mesoporous Phosphorus-Doped Alumina Supported Palladium Catalysts for Methane Combustion. ACS Catal. 2018, 8, 11016–11028.

l  Lundgren, E.; Nilsson, J.; Gustafson, J.; Grönbeck, H.; Skoglundh, M.; Newton, M.A.; Fouladvand, S.; Carlsson, P.-A.; Martin, N.M. Chemistry of Supported Palladium Nanoparticles during Methane Oxidation. ACS Catal. 2015, 5, 2481–2489.

l  Gorte, R.J.; Arroyo-Ramirez, L.; Chung, Y.-C.; Zhang, S.; Graham, G.W.; Onn, T.M.; Pan, X. Improved Thermal Stability and Methane-Oxidation Activity of Pd/Al 2 O 3 Catalysts by Atomic Layer Deposition of ZrO2 . ACS Catal. 2015, 5, 5696–5701.

l  Goodman, E.D.; Dai, S.; Yang, A.C.; Wrasman, C.J.; Gallo, A.; Bare, S.R.; Hoffman, A.S.; Jaramillo, T.F.; Graham, G.W.; Pan, X.; et al. Uniform Pt/Pd Bimetallic Nanocrystals Demonstrate Platinum Effect on Palladium Methane Combustion Activity and Stability. ACS Catal. 2017, 7, 4372–4380.

l  Cargnello, M.; Jaén, J.J.D.; Garrido, J.C.H.; Bakhmutsky, K.; Montini, T.; Gámez, J.J.C.; Gorte, R.J.; Fornasiero, P. Exceptional Activity for Methane Combustion over Modular Pd@CeO2 Subunits on Functionalized Al2O3. Science 2012, 337, 713–718.

l  Pantaleo, G.; Liotta, L.F.; Venezia, A.M.; Kantcheva, M.; Di Carlo, G. Effect of Ti(IV) loading on CH4 oxidation activity and SO2 tolerance of Pd catalysts supported on silica SBA-15 and HMS. Appl. Catal. B Environ. 2011, 106, 529–539.

l  Liotta, L.F.; Carlo, G. Di; Pantaleo, G.; Garrido, J.C.H.; Venezia, A.M. Pd(1 wt%)/LaMn0.4Fe0.6O3 catalysts supported over silica SBA-15: Effect of perovskite loading and support morphology on methane oxidation activity and SO2 tolerance. Top. Catal. 2012, 55, 782–791.

Response 1: Relevant literatures have been supplemented in the reference list as below.

[3] Chen, J.; Arandiyan, H.; Gao, X.; Li, J. Recent Advances in Catalysts for Methane Combustion. Catal. Surv. Asia. 2015, 140–171.

[5] Ji, Y.; Guo, Y. B. Nanostructured perovskite oxides as promising substitutes of noble metals catalysts for catalytic combustion of methane. Chin. Chem. Lett. 2018, 29, 252-260.

[7] Monai, M.; Montini, T.; Gorte, R. J.; Fornasiero, P. Catalytic Oxidation of Methane: Pd and Beyond. Eur. J. Inorg. Chem. 2018, 25, 2884–2893.

[8] Jiang, L.; Zheng, Y.; Chen, X.; Xiao, Y.; Cai, G.; Zheng, Y.; Zhang, Y.; Huang, F. Catalytic Activity and Stability over Nanorod-Like Ordered Mesoporous Phosphorus-Doped Alumina Supported Palladium Catalysts for Methane Combustion. ACS Catal. 2018, 8, 11016–11028.

[22] Venezia A. M.; Carlo G. D.; Liotta L. F.; Pantaleo G.; Kantcheva M. Effect of Ti(IV) loading on CH4 oxidation activity and SO2 tolerance of Pd catalysts supported on silica SBA-15 and HMS. Appl. Catal. B. 2011, 106, 529-539.

[48] Banerjee, A. C., Golub, K. W., Abdul, Md. H., Billor, M. Z. Comparative study of the characteristics and activities of Pd//γ-Al2O3 catalysts prepared by Vortex and Incipient Wetness Methods. Catalysts, 2019, 9, 336.

[49] Banerjee, A. C., McGuire, M. M., Lawnick, O., Bozack, M. J. Low-temperature activity and PdO/PdOx transition in methane combustion by a PdO-PdOx/y-Al2O3 catalyst. Catalysts, 2018, 8, 266.

[50] Cargnello, M.; Jaén, J. J. D.; Garrido, J. C. H.; Bakhmutsky, K.; Montini, T.; Gámez, J. J. C.; Gorte, R. J.; Fornasiero, P. Exceptional Activity for Methane Combustion over Modular Pd@CeO2 Subunits on Functionalized Al2O3. Science, 2012, 337, 713–718.

2. Line 49: Revise the sentence “To the best of our knowledge, in the literature no sulfur-tolerant palladium system has been used for the catalytic oxidation of methane” and incorporate references on sulfur-tolerant Pd catalyst reported in some of the articles mentioned above.

Response 2: The sentence has been revised as “The research on sulfur-tolerant palladium catalyst system in methane catalytic oxidation reaction is rare and lack of substantial progress in the literature [18, 24-26 ].”

[18] Wilburn, M. S.; Epling, W. S. Sulfur deactivation and regeneration of mono- and bimetallic Pd-Pt methane oxidation catalysts. Appl. Catal. B. 2017, 206, 589-598

[24] Meeyoo, V.; Trimm, D. L.; Cant, N. W. The effect of sulphur containing pollutants on the oxidation activity of precious metals used in vehicle exhaust catalysts. Appl. Catal. B. 1998, 16, L101–L104.

[25] Ordóñez, S.; Hurtado, P.; Díez, F. V. Methane catalytic combustion over Pd/Al2O3 in presence of sulphur dioxide: development of a regeneration procedure. Catal. Lett. 2005, 100, 27-34.

[26] Corro, G.; Cano, C.; Fierro, J. L. G. A study of Pt–Pd/γ-Al2O3 catalysts for methane oxidation resistant to deactivation by sulfur poisoning. J. Mol. Catal. A: Chem 2010, 315, 35-42.

3. Introduction: Explain why H2S was chosen to test sulfur-resistance and why not SO2. Explain how in real life conditions, H2S will be part of a system where such catalyst could be used and then relate how this study could have practical applications.

Response 3: As SO2 is easy to oxidize and dissolve in water, H2S is the main sulfur impurity in natural gas. The selection of H2S to test sulfur-resistance is more consistent with the actual composition of natural gas. This is also consistent with relevant literatures [24-26]. Effective catalytic oxidation of CH4 can improve combustion efficiency and reduce air pollutants, such as CO, NOx, and unburned hydrocarbon [3-8]. It has therefore found great applications in modern industry, such as catalytic exhaust converters aimed to reduce methane emission and catalytic gas turbine combustors designed to combust fuel under mild conditions [9, 10].

[3] Chen, J.; Arandiyan, H.; Gao, X.; Li, J. Recent Advances in Catalysts for Methane Combustion. Catal. Surv. Asia. 2015, 140–171.

[4] Colussi, S.; Gayen, A.; Camellone, M. F.; Boaro, M.; Llorca, J.; Fabris, S; Trovarelli, A. Nanofaceted Pd-O Sites in Pd-Ce Surface Superstructures: Enhanced Activity in Catalytic Combustion of Methane. Angew.  Chem. 2009, 121,8633-8636.

[5] Ji, Y.; Guo, Y. B. Nanostructured perovskite oxides as promising substitutes of noble metals catalysts for catalytic combustion of methane. Chin. Chem. Lett. 2018, 29, 252-260.

[6] Bossche, M. V. D.; Gronbeck, H. Methane Oxidation over PdO(101) Revealed by First-Principles Kinetic Modeling. J. Am. Chem. Soc. 2015, 137, 12035-12044.

[7] Monai, M.; Montini, T.; Gorte, R. J.; Fornasiero, P. Catalytic Oxidation of Methane: Pd and Beyond. Eur. J. Inorg. Chem. 2018, 25, 2884–2893.

[8] Jiang, L.; Zheng, Y.; Chen, X.; Xiao, Y.; Cai, G.; Zheng, Y.; Zhang, Y.; Huang, F. Catalytic Activity and Stability over Nanorod-Like Ordered Mesoporous Phosphorus-Doped Alumina Supported Palladium Catalysts for Methane Combustion. ACS Catal. 2018, 8, 11016–11028.

[9] Fouladvand, S.; Skoglundh, M.; Carlsson, P. A. A transient in situ infrared spectroscopy study on methane oxidation over supported Pt catalysts. Catal. Sci. Technol. 2014, 4, 3463-3473.

[10] Florén, C. R.; Bossche, M. V. D.; Creaser, D.; Grobeck, H.; Carlsson, P. A.; Korpi, H.; Skoglundh, M. Modelling complete methane oxidation over palladium oxide in a porous catalyst using first-principles surface kinetics. Catal. Sci. Technol. 2018, 8, 508-520.

[24 ]Meeyoo, V.; Trimm, D. L.; Cant, N. W. The effect of sulphur containing pollutants on the oxidation activity of precious metals used in vehicle exhaust catalysts. Appl. Catal. B. 1998, 16, L101–L104.

[25] Ordóñez, S.; Hurtado, P.; Díez, F. V. Methane catalytic combustion over Pd/Al2O3 in presence of sulphur dioxide: development of a regeneration procedure. Catal. Lett. 2005, 100, 27-34.

[26] Corro, G.; Cano, C.; Fierro, J. L. G. A study of Pt–Pd/γ-Al2O3 catalysts for methane oxidation resistant to deactivation by sulfur poisoning. J. Mol. Catal. A: Chem 2010, 315, 35-42.

4.  2. 1. Results and discussion

2.1. Fresh catalysts

 Add sub-topics : Preparation, characterization and activity under 2.1

Response 4: Relevant sub-topics have been supplemented.

 Figure 1 and Table 1 contains catalysts with abbreviations (f 0.2-f10). Define these in the text when such abbreviations first appear; also define FWHM;

Response 5: Relevant definitions have been supplemented in the text as below:

The freshly prepared catalyst is named (fV), where V represents H2S concentration.

FWHM: full-width at half-maximum

 State the technique/ the method used to determine the particle sizes and add some details to explain the data

Response 6: Relevant explain and some details of the method used to determine the particle sizes have been supplemented in the text as below:

Average particle size was determined from XRD measurements using the Scherrer formula: . Where d is the average particle size (nm), K is the Scherrer constant and the diffraction angle is denoted θ, λ = 0.154 nm stands for the wavelength of Cu K-a radiation, β denotes the full-width at half-maximum (FWHM) of diffraction peak.

Lines 85-91: Explain HRTEM Figure 2 and results more in details.

Response 7: Relevant explain and some details of the method used to determine the particle sizes have been supplemented in the text as below:

The lattice spacing distance of Pd species particles on freshly prepared catalyst are 0.225 nm and 0.245 nm, corresponding to the Pd(111) and Pd4S(102), respectively [JCPDS No. 73-1387, JCPDS No. 46-1043].

Activity: Mention inlet gas composition, particle size, flow rate, gas hourly space velocity in the text and all tables on activities

Response 8: Relevant conditions have been supplemented in the text and all tables on activities as below:

Gas composition: CH4 (v% = 2 %), O2 (v% = 8 %), H2S (v% = 0.02 %), N2 (v% = 89.8%); Flow rate = 200 mL/min; GHSV = 60000 ml/g.h

 Suggestion on Figure 4: This figure contains too much of data. Make one figure on conversion vs temp and another on conversion vs reaction time

Response 9: The catalytic oxidation of methane has its own characteristics. When the reaction temperature was 500 and 370 , all the catalysts exhibit almost the same catalytic performance. Only when the reaction temperature was 400 , the sulfur-tolerance and activity of different catalysts could be compared effectively. Therefore, it is not suitable to divide Figure 4 into two figures.

2. 2 Used catalysts

Please address the points mentioned for fresh catalysts here also

Response 10: Relevant points mentioned for used catalysts have been supplemented in the text as below:

Relevant sub-topics have been supplemented.

The used catalyst is named (uV), where V represents H2S concentration.

Gas composition: CH4 (v% = 2 %), O2 (v% = 8 %), H2S (v% = 0.02 %), N2 (v% = 89.8%); Flow rate = 200 mL/min; GHSV = 60000 ml/g.h

5. 3. Materials and Methods

3. 1. Catalyst preparation: Include more details about each technique and methods for catalyst preparation

Response 11: Relevant details have been supplemented in catalyst preparation as below:

The PdO/SiO2 catalyst was prepared by isovolumetric impregnation. Typically, weigh a proper quantity of 40~60 mesh SiO2 (M = 10 g Vg = 0.94 cm3/g, Rd = 10.6 nm, S = 353.0028 m2/g, Qingdao Baisha Catalyst Factory) was dispersed in palladium acetate (Aladdin Industrial Corporation) aqueous solution (VPd = 0.05 g/mL) overnight. Then the sample was dried at 110 oC for 4 h and calcined at 500 oC for 4 h in the air. The theoretical loading of Pd was 5 wt%.

The PdO/SiO2 was first reduced to Pd/SiO2 by H2 at 500 oC for 1h. PdxSy/SiO2 catalyst was prepared using above Pd/SiO2 as precursor. 2.0 g Pd/SiO2 and 100 ml deionized water were stirred at 350 rpm in a three-necked flask. Adjust the flow rate of H2S and H2 by flow controller (China, Beijing Sevenstart Electronics Co.Ltd, D07-19B) to configure different concentrations of H2S/H2 mixture. Then 30 mL/min of H2S-H2 (VH2S% = 0.2, 1, 3, 5, 7, 10) was fed into the suspension at 60 °C for 1 h. Afterwards, the sample was filtered and rinsed with distilled water till neutral. Finally, the sample was dried at 110 oC for 4 h. The freshly prepared catalyst is named (fV), and the used catalyst is named (uV), where V represents H2S concentration.

3. 2 Describe details about characterization techniques and give references; explain how compositions by XPS were determined (Table 3)

Response 12: Relevant details have been supplemented as below:

The XPS profiles were fitted by the software namedXPS peak. Then the fitting peak area was used to calculate the proportion of different species.

3.3 Evaluation of the catalysts: include more details including how temperature was controlled and how flow rate was measured. Mention the company, state and country for all equipment and chemicals, gas mixture as a requirement for this journal.

Response 13: Relevant details have been supplemented as below:

Gases consisting of CH4 (v% = 2 %), O2 (v% = 8 %), H2S (v% = 0.02 %), and N2 were fed through the flow controllerChina, Beijing Sevenstart Electronics Co.Ltd, D07-19Bat 200 mL/min and used in all the experiments. Finally, the temperature controller (China, XiaMen YuDian Automation Technology Co.,Ltd, AI808PK1L2) is used to control different reaction temperatures. The effluent gases were periodically sampled and simultaneously analyzed online by gas chromatographs (China, Fuli, GC-9790).

6. Suggested additional experiments:

The authors claim that the catalyst they developed is sulfur resistant. That is a good progress. However, any good catalysts must pass the following tests and these must be done and reported in the revised manuscript:

(a) 72-hour stability test without any sulfur-containing component

(b) 72-hour stability test with a gas mixture having sulfur-containing component

(c) Stability of the catalyst in the presence of steam

Response 14: Relevant 72-hour stability tests with and without H2S have been supplemented. as Figure 11. It could be found that the catalyst could maintain long-term stability regardless of the presence of H2S in the reaction atmosphere. Stability of the catalyst in the presence of steam needs to carry out large-scale revamping of the catalyst evaluation units. Therefore, the study of the effect of steam will be carried out in the future.

Figure 11. Long-term stability test of the u7 catalyst with and without H2S. Gas composition: CH4 (v% = 2 %), O2 (v% = 8 %), H2S (v% = 0.02 %), N2 (v% = 89.8%); Flow rate: 200 mL/min; GHSV=60000 ml/g.h.

Reviewer 2 Report

The paper "Pd4S/SiO2: A sulfur-tolerant palladium catalyst for catalytic oxidation of methane" (Manuscript Number: catalysts-489207) is devoted to the catalytic oxidation of methane using a sulfur-tolerant catalyst (Pd4S/SiO2). The prepared catalysts were characterized by X-ray diffraction, HR-TEM, XPS. The oxidation stability of the catalysts was investigated by recording the mass change in the process of air oxidation. Also, temperature-programmed desorption (TPD) of hydrogen sulfide experiments were performed. In general, the studies of sulfur-tolerant catalysts for catalytic oxidation of hydrocarbons are very important from the point of view of modern industrial chemistry.

However, the composition of methane oxidation products is not described. Conversion of oxidation products should be represented along with methane conversion.

I think that the manuscript may be published after a minor improvement.

Author Response

Dear Editor:

We would like to thank the editor for giving us a chance to revise the paper and the reviewers for their constructive suggestions to improve the quality of the paper. Here we submit a revision with the title Pd4S/SiO2: A sulfur-tolerant palladium catalyst for catalytic complete oxidation of methane” (ID: Catalysts-489207), which has been modified according to the reviewers’ suggestions. All detailed revisions are listed in response to comments. We hope this version can satisfy the quality of Catalysts.

Sincerely yours,

Lei Ma

--------------------------------------------------------------------------------------------------

The following is a point-to-point response to the reviewers’ comments.

REVIEWER REPORT(S):

Referee: 2

The paper "Pd4S/SiO2: A sulfur-tolerant palladium catalyst for catalytic oxidation of methane" (Manuscript Number: catalysts-489207) is devoted to the catalytic oxidation of methane using a sulfur-tolerant catalyst (Pd4S/SiO2). The prepared catalysts were characterized by X-ray diffraction, HR-TEM, XPS. The oxidation stability of the catalysts was investigated by recording the mass change in the process of air oxidation. Also, temperature-programmed desorption (TPD) of hydrogen sulfide experiments were performed. In general, the studies of sulfur-tolerant catalysts for catalytic oxidation of hydrocarbons are very important from the point of view of modern industrial chemistry.

However, the composition of methane oxidation products is not described. Conversion of oxidation products should be represented along with methane conversion.

I think that the manuscript may be published after a minor improvement.

Response 1: The reaction studied in this paper is catalytic complete oxidation of methane (CH4 + 2O2 → CO2 + 2H2O). Therefore, oxidation products are carbon dioxide and water. Complete oxidation has been added to the title and related positions of the article by adding "complete" description.

Round 2

Reviewer 1 Report

Comments on the revised manuscript included in the attachment 

Author Response

Comments on the revised manuscript Catalysts-489207

I read the responses given by the authors in response to my first round of reviews. The authors have addressed most of the suggestions and comments in the revised manuscript. The revised manuscript is much better compared to the first submission. However, I suggest the following additional revisions to improve the manuscript further.

1. Placement of citations in the text:

The following references are currently wrong place. I suggest the authors consider citing these references at the end of this sentence in the revised manuscript under introduction; lines 34-35) : “Supported PdO catalysts have shown excellent catalytic property in methane oxidation, and currently are under extensive study [ ]”

48. Banerjee, A. C., Golub, K. W., Abdul, Md. H., Billor, M. Z. Comparative study of the characteristics and 465 activities of Pd//γ-Al2O3 catalysts prepared by Vortex and Incipient Wetness Methods. Catalysts, 2019, 9, 466 336.

49. Banerjee, A. C., McGuire, M. M., Lawnick, O., Bozack, M. J. Low-temperature activity and PdO/PdOx 468 transition in methane combustion by a PdO-PdOx/y-Al2O3 catalyst. Catalysts, 2018, 8, 266.

Response 1: The two references have been cited in the correct place at the end of “Supported PdO catalysts have shown excellent catalytic property in methane oxidation, and currently are under extensive study [11-18]”, lines 37-38

[11] Banerjee, A. C., Golub, K. W., Abdul, Md. H., Billor, M. Z. Comparative study of the characteristics and activities of Pd/γ-Al2O3 catalysts prepared by Vortex and Incipient Wetness Methods. Catalysts, 2019, 9, 336.

[12] Banerjee, A. C., McGuire, M. M., Lawnick, O., Bozack, M. J. Low-temperature activity and PdO/PdOx transition in methane combustion by a PdO-PdOx/y-Al2O3 catalyst. Catalysts, 2018, 8, 266

2. Abstract

The abstract does not include some of the important features the authors discussed in the body of the manuscript. The following new areas/ revisions should be added to the abstract and the abstract may be rewritten to reflect the text of the manuscript. Editing are marked yellow.

A. The sentence “The previous methods ---- species” may be rewritten as: “The previously reported methods of improving sulfur-tolerance were effective to defer the deactivation of palladium catalysts, but could not prevent PdO and carrier materials from interacting with sulfur species.”

B. Revise to add : “ In this report, new sulfur-tolerant SiO2 supported Pd4S catalysts ( --Pd wt% loading) were prepared by H2S-H2 17 aqueous bubble method and applied to catalytic complete oxidation of methane.”

C. Add the characterization techniques: The catalysts were characterized by -------- techniques [ Do not use abbreviations]

D. Revise and Add: The ----catalyst was effective to convert --% methane under the conditions --- and also exhibited long-term stability.

E. Revise the sentence: “Thus, the first Pd4S/SiO2 24 catalyst that possesses excellent sulfur-tolerance, oxidation stability and catalytic activity has been 25 developed for catalytic complete oxidation of methane.” To : “ A Pd4S/SiO2 catalyst that possesses excellent sulfur-tolerance, oxidation stability and catalytic activity has been 25 developed for catalytic complete oxidation of methane.”

Response 2: The abstract has been supplemented and revised as recommended, as follows:

Abstract: Sulfur species (e.g. H2S or SO2) are the natural enemies of most metal catalysts, especially palladium catalysts. The previously reported methods of improving sulfur-tolerance were effectively defer the deactivation of palladium catalysts, but could not prevent PdO and carrier interacting between sulfur species. In this report, novel sulfur-tolerant SiO2 supported Pd4S catalysts (5 wt% Pd loading) were prepared by H2S-H2 aqueous bubble method and applied to catalytic complete oxidation of methane. The catalysts were characterization by X-ray diffraction, transmission electron microscopy, X-ray photoelectron Spectroscopy, temperature-programmed oxidation and temperature-programmed desorption techniques under identical conditions. The structural characterization revealed that Pd4S and metallic Pd0 were found on the surface of freshly prepared catalysts. However, Pd4S remained stable while most of metallic Pd0 was converted to PdO during the oxidation reaction. When coexisting with PdO, Pd4S not only protected PdO from sulfur poisoning, but also determined the catalytic activity. Moreover, the content of Pd4S could be adjusted by changing H2S concentration of H2S-H2 mixture. When H2S concentration was 7 %, the Pd4S/SiO2 catalyst was effective to convert 96% methane at 400 oC and also exhibited long-term stability in the presence of 200 ppm H2S. A Pd4S/SiO2 catalyst that possesses excellent sulfur-tolerance, oxidation stability and catalytic activity has been developed for catalytic complete oxidation of methane.

3. English Language and Grammar: Needs extensive editing

Response 3: English language and grammar have been optimized and edited.
